# Enhancing Strength and Ductility of a Ni-26.6Co-18.4Cr-4.1Mo-2.3Al-0.3Ti-5.4Nb Alloy via Nanosized Precipitations, Stacking Faults, and Nanotwins

**DOI:** 10.3390/nano14151296

**Published:** 2024-07-31

**Authors:** Jingjing Zhang, Yongfeng Shen, Wenying Xue, Zhijian Fan

**Affiliations:** 1Key Laboratory for Anisotropy and Texture of Materials, Ministry of Education, School of Materials Science and Engineering, Northeastern University, Shenyang 110819, China; jjzhang1995@163.com; 2The State Key Laboratory of Rolling & Automation, Northeastern University, Shenyang 110819, China; xuewy@ral.neu.edu.cn; 3Key Laboratory of Neutron Physics and Institute of Nuclear Physics and Chemistry, China Academy of Engineering Physics, Mianyang 621999, China; fanzhijian@caep.cn

**Keywords:** Ni-based alloy, annealing time, mechanical properties, precipitation strengthening, deformation mechanism

## Abstract

The addition of Co to Ni-based alloys can reduce the stacking fault energy. In this study, a novel Ni-26.6Co-18.4Cr-4.1Mo-2.3Al-0.3Ti-5.4Nb alloy was developed by increasing the Co addition to 26.6 wt.%. A new strategy to break the trade-off between strength and ductility is proposed by introducing dense nanosized precipitations, stacking faults, and nanoscale twins in the as-prepared alloys. The typical characteristics of the deformed alloy include dense dislocations tangles, nanotwins, stacking faults, and Lomer–Cottrell locks. In addition to the pinning effect of the bulky δ precipitates to the grain boundaries, the nanosized γ′ particles with a coherent interface with the matrix show significant precipitation strengthening. As a result, the alloy exhibits a superior combination of yield strength of 1093 MPa and ductility of 29%. At 700 °C, the alloy has a high yield strength of 833 MPa and an ultimate tensile strength of 1024 MPa, while retaining a ductility of 6.3%.

## 1. Introduction

Ni-based superalloys have extensive applications in industry. With the advancement of industrial technology, higher requirements have been put forward for the comprehensive performance of Ni-based superalloys. For example, many applications require higher temperature resistance and mechanical properties of Ni-based superalloys [1,2].

Currently, Inconel 718 is the most widely used alloy in aero-engines because of its excellent strength and manufacturability [3,4,5]. However, the γ″ phase in the alloy is unstable and tends to evolve into a δ phase when the alloy is kept at high temperatures for a long duration. This results in a sharp decrease in the mechanical strength of Inconel 718, thereby limiting its use at temperatures around 650 °C. Therefore, modulation of the alloying elements to reduce unstable strengthening phases and the rational use of structural defects are effective ways to improve the mechanical properties of such superalloys.

Recently, γ′-strengthened superalloys have attracted great interest in high-temperature applications because of their excellent high-temperature strength which is derived from the γ′ precipitation (L1_2_ structure) [6,7]. Kümmel et al. [8] studied the deformation mechanism of differently oriented grains in the VDM 780 alloy. The results suggested that dislocation motion and shearing of γ′ precipitation played key roles during the deformation that was strongly anisotropic. Sun et al. [9] prepared nanoscale (*d* = 50 nm) Ni-based alloys using a rolling process, which showed that γ’ precipitation contributed to ductility and thermal stability. Zhang et al. reported a novel L1_2_-strengthening NiCoCrAlTi high-entropy alloy (HEA) with an outstanding combination of tensile strength and ductility—the yield stress and ductility of the HEA were ~1300 MPa and 14% at room temperature, respectively [10].

The deformation mechanisms and mechanical properties of *fcc* metals are strongly related to their stacking faults energy (SFE, γ_SFE_), which is an important indicator determining whether twinning, martensite transformation, and dislocation glide will dominate the deformation process of the material. For materials with low SFE, the plastic deformation mode may change from dislocation slip to deformation twinning, which is very important for material strengthening [11,12]. In addition, it has been reported that the deformation mechanisms of Ni-based superalloys vary with reduction in SFE by Co addition [13,14]. Ni-based superalloys with low SFE are conducive to forming dense SFs and deformation twins during plastic deformation [15]. Yang et al. [16] developed a novel Ni-based superalloy with 20 wt.% Co to reduce SFE, as a result, the yield strength of the alloy with 6% pre-tensile deformation increased by 29.8%, and the stress rupture life was maintained at 74.1% at 760 °C/480 MPa. This was attributed to the formation of numerous dislocations, slip bands, and SFs during the deformation of the alloy. Wang et al. successfully prepared nano-twin crystals by cold-rolling and found the alloy was stable even after annealing at 950 °C for 2 h [17]. To continuously slip, the perfect dislocation tends to dissociate into two partial dislocations separated by an SF when it encounters an obstacle, especially for metals with low SFE, such as Ag, Cu, and nickel-based alloys [18]. On the one hand, the occurrence of partial dislocations promotes the twinning behavior because the SF serves as a twin nucleus based on the models of defect-assisted nucleation [19,20]. On the other hand, the low SFE enhances planar glide and hinders cross-slip [21,22]. That is to say, the SFE affects both strength and ductility. Deformation twins should be activated during tensile test provided with the low SFE. Thus, both the work hardening rate and ductility are increased [23].

In this study, we develop a novel Ni-based superalloy with a high Co content of 26.6 wt.%, which aims to decrease the stacking fault energy. Thus, increasing strain hardening is expected to optimize the combination of strength and ductility of Ni-based alloys, depending on the formation of dense stacking faults and nanotwins. 

## 2. Materials and Methods

An alloy with a chemical composition of Ni-26.6Co-18.4Cr-4.1Mo-2.3Al-0.3Ti-5.4Nb (in wt. %), measured by inductively coupled plasma mass spectroscopy, was used in this study. The alloy was melted in a vacuum furnace and then a 380 × 96 × 38 mm^3^ ingot was cast. The ingot was solution-treated (named STed) in an air furnace at 1200 °C for 2 h to homogenize the microstructure. Subsequently, the resultant ingot was hot-rolled at a starting temperature of 1200 °C for five passes to obtain a plate with a thickness from 18 mm to 4.5 mm. The final rolling temperature was 880 °C, which was measured by using a UT300S infrared temperature measuring gun, after which the plate was cooled in air to room temperature. The surfaces of the hot-rolled plate were polished to eliminate the oxide layers, and then the plate was rolled to a thickness of 2 mm at room temperature (RT). Finally, the plate was divided into 3 parts and then annealed at 1000 °C for different durations (10 min, 30 min, 60 min), respectively. Then, the three plates were aged at 720 °C for 8 h followed by cooling to 620 °C at a rate of 50 °C/h in the furnace. Subsequently, the plates were aged at 620 °C for 8 h followed by air cooling (AC) to RT. The resultant plates were named A10A, A30A, A60A, respectively, and the corresponding treatment processes are shown in Figure 1a.

To measure the mechanical properties, dog-bone shaped specimens with a gage dimension of 18 × 5 × 1.5 mm^3^ were cut along the rolling direction (RD) by using wire electrical discharge machining (Figure 1b). Tensile tests were carried out on an AG-X plus 100 kN (Shimadzu, Kyoto, Japan) testing machine at a strain rate of 1 × 10^−3^ s^−1^. To guarantee the precision of the data obtained, three tensile specimens treated at the identical heat treatment process were used to obtain average results. 

The specimens used for X-ray diffraction (XRD, Cu Kα radiation) testing were gradually ground to 2000# with SiC sandpaper (Jiangsu, China). The scanning range was 20° and 120° (2*θ*) at a step of 0.03°/s and a counting time of 0.6 s per step. The microstructures of the alloys were characterized using a JSM-7001F scanning electron microscope (SEM, Tokyo, Japan). Sheets of 5 × 10 × 1.5 mm^3^ were cut along the RD from the heat-treated plates. The samples were electrolytically polished under an electrolyte consisting of 10% perchloric acid and 90% alcohol for SEM and EBSD observation. The SEM observation voltage was 20 kV. Electron-backscatter diffraction (EBSD) analyses were performed on a scanning electron microscope (JSM7001F, Tokyo, Japan) equipped with an EBSD detector. EBSD data and images were processed using HKL Channel 5 software.

Transmission electron microscope (TEM) and selected area electron diffraction (SAED) studies were conducted using a JEM2100F (Tokyo, Japan) field emission electron microscope operating at 200 kV to provide the microstructural features in detail. The observed sections of the samples for TEM were in the ND–TD plane. Disks with a diameter of 3 mm cut from the plane were mechanically ground to a thickness of 40–50 μm. Finally, the discs were thinned in a solution of 10% perchloric and 90% alcohol solution by using a twin-jet electro-polisher operated at 32 V and −25 °C. RD, TD, and ND stand for rolling, transverse, and normal directions, respectively.

## 3. Results

### 3.1. Microstructures

The SEM images show that numerous fine particles with strip-like morphology locate at the grain boundaries that are parallel to the RD in all the three samples (Figure 2a–c). Meanwhile, it is evident that the numbers of strip-like precipitates gradually decrease with increase in annealing time, indicated by yellow arrows. This is related to the re-dissolution of the precipitates during annealing at high temperatures. The inverse pole figure (IPF) maps exhibit that the grain sizes vary significantly with the annealing temperatures, and no obvious crystallographic texture can be seen in the three samples (Figure 2(a1)–(c1)). The statistical results show that the average grain size of the samples gradually increases from 7.4 μm to 10.8 μm and 15.2 μm with increase in the annealing durations (Figure 2(a2)–(c2)). It should be pointed out that a lot of small grains locate at the vicinity of the strip-like precipitates. This is reasonable because the precipitates can block dislocation movements and the pinned grain boundaries, hence retarding the recrystallization [24,25]. In addition, Figure 2(a3)–(c3) show the related orientation distribution function (ODF) sections with φ_2_ = 45° for the alloys A10A to A60A. Since all alloys have the complete fcc structure, the ideal orientation in the ODF sections of the Euler space [26] are shown in Figure 2d for reference. Obviously, the microstructure in the aged alloys is clearly selectively oriented, i.e., there are strong texture features. Specifically, the A10A and A30A alloys show a strong near-brass deformation texture ({110} <112>), while the A60A alloy exhibits a strong Goss recrystallization texture ({110} <001>). All three alloys show a weak near-copper texture ({112} <111>). This indicates that the degree of recrystallization of the alloy increases with the annealing time.

Based on the fact that a large number of annealed twins with the Σ3 boundaries were observed in the three specimens (Figure 2(a1)–(a3)), close observations of the twins in detail were conducted. The EBSD IPF maps (Figure 3a–c) and the corresponding grain boundaries maps (Figure 3(a1)–(c1)) were obtained. Subsequently, the area fraction of twins and the percentage of twin boundaries in the total grain boundaries were evaluated. The statistical results show that the areas of the twins increased from 14% to 19%, while the ratio of twin boundaries to total boundaries increased from 46% to 53% with increase in annealing time for the three specimens. The results suggest that the increasing annealing durations are conducive to the formation of twins.

Figure 4a shows the XRD patterns of the A10A, A30A, A60A samples, compared with the STed specimen. Evidently, all the diffraction peaks of the STed sample are consistent with those of the pure nickel standard (JCPDS card 01-087-0712), confirming that the alloy consists entirely of the γ matrix. For the annealed specimens, the (200), (220), (311), and (222) peaks gradually become weak while those of the (111) peaks appear to become strong with increase in the annealing duration. This indicates that the number of grains oriented in (111) decreases significantly, which means that the alloys have developed a preferential orientation during the heat treatment (as shown in Figure 2(a3)–(c3)). The magnified sections of the (111) peaks in Figure 4a show that the extra minor peaks are detected in the vicinity of each peak, which can be identified as the γ′ phase (Figure 4b). In addition, compared with the Sted alloy, a shift in the (111) diffraction peak of the aged alloys (A10A, A30A, A60A) is observed in Figure 4b, which indicates that the lattice constants of the alloys become smaller after aging treatment. The phenomenon is caused by the precipitation of γ′ on the matrix. No other phases can be detected; hence, the features of the strip precipitates (Figure 2) still require further characterization in detail.

Systematic TEM observations were conducted and the typical results obtained are presented in Figure 5, showing the morphologies of the two typical precipitates. Several large particles can be observed along the grain boundaries, with an average size of ~200 nm, and the corresponding energy dispersive spectroscopy (EDS) maps show that the precipitates are mainly enriched in Ni, Nb, and Mo while being reduced in Cr (Figure 5a). Interestingly, the aggregation of the particles forms a child-like shape, with the left arm rising. To verify the characteristics of the large particles, one oblong particle along the boundary (Figure 5b) was selected to be observed by the selected area electron diffraction (SAED). The obtained SAED pattern proves that this precipitate is the δ phase (Ni_3_Nb) due to the crystal plane spacing of 0.194 nm for the (110) plane (Figure 5c), which is consistent with observations for the Inconel 718 [27] and VDM 780 Ni-based superalloys [28]. The results suggest that sufficient pre-precipitation and uniform dispersion of the δ phase play key roles in achieving significant grain refinement. In this study, the δ phase can generally be observed along the grain boundaries, suggesting a similar effect of the particles during the rolling processing. 

On the other hand, numerous nanosized particles are observed, which have sizes of several nanometers, and are widely distributed in the matrix (Figure 5d,e). The size of these precipitates is approximately constant due to the identical aging process for the three samples. According to the statistical results for seven close views identical to Figure 5e, the average size of the precipitates is 10 ± 2 nm, and the corresponding SAED patterns show that these precipitates are γ′ phase. In addition, the γ′ phase and the matrix follow the misorientation relationships of [100]_γ′_//[110]_γ_. 

### 3.2. Mechanical Properties

Figure 6a shows the engineering stress–strain curves of the three Ni-26.6Co-18.4Cr-4.1Mo-2.3Al-0.3Ti-5.4Nb alloys tested at room temperature. As can be seen, the as-prepared alloys exhibit concurrently increasing strength and ductility, compared with the STed specimen. For example, the yield and ultimate tensile strengths of the STed sample are only 575 MPa and 828 MPa, respectively, with a total elongation of 23%. In contrast, the A10A alloy shows substantially high yield and ultimate tensile strengths of 1029 MPa and 1480 MPa. Beyond expectation, the ductility slightly increases to 27%, which is encouraging because a trade-off between the strength and ductility is usually unavoidable for metallic materials. In general, the strength of the metals increases significantly due to work hardening induced by the rolling processing, while the elongation decreases [29]. In particular, the yield and ultimate tensile strengths further increase to 1093 MPa and 1561 MPa when the annealing time increases from 10 min to 30 min, accompanied with a slight increase in ductility from 27% to 29%. When the annealing time further increases to 60 min, the yield and ultimate tensile strengths unfavorably decrease to 988 MPa and 1417 MPa, respectively, despite the weak increase in ductility (30%). Evidently, by adopting the rolling and annealing treatments, not only the yield and ultimate tensile strengths but also the ductility of the resultant alloys simultaneously increase compared to the STed sample. The results show that the optimal annealing duration is 30 min for the Ni-26.6Co-18.4Cr-4.1Mo-2.3Al-0.3Ti-5.4Nb alloy to achieve a superior combination of strength and ductility at room temperature. 

Figure 6b shows the engineering stress and strain curves of the three samples tested at 700 °C. The strengths of the specimens significantly decrease by about 200 MPa in comparison with the values tested at room temperature. Nevertheless, it should be pointed out that the strengths are still high at such a high temperature. Among the three alloys, the A30A alloy exhibits the highest yield and ultimate tensile strengths of 833 MPa and 1024 MPa, while retaining a good ductility of 6.3%. By contrast, the A10A alloy has the lowest yield and ultimate tensile strengths of 813 MPa and 975 MPa, with a low ductility of 4.9%. It has been reported that the VDM 780 alloy has a yield strength of 690 MPa at 649 °C, with a ductility of ~5% [8]. Hence, the A30A alloy has the distinct advantages of strength and ductility in comparison with the reported alloy at the elevated temperature. Meanwhile, one can notice that the ductility dramatically drops for the three samples at 700 °C. This should be mainly related to the occurrence of mid-temperature embrittlement fracture during the process, which refers to the decrease in ductility of superalloys at mid-temperature, generally caused by the violent thermal vibration of atoms at high temperatures [30]. It is conceivable that this weakens the strength of metallic bonds, thus leading to grain-boundary segregation. Consequently, the local deformation is enhanced, and the homogenous deformation is retarded accordingly. 

Overall, the A30A sample shows good mechanical properties at room and high temperatures, with excellent mechanical properties at room temperature. Hence, the sample fractured at room temperature is selected for exploring the deformation mechanisms in the next section. 

## 4. Discussion

### 4.1. Microstructural Evolution

Based on SEM and TEM observations, the microstructural evolutions of the alloys during the preparation are shown in schematic diagrams (Figure 7). It is found that cold-rolling promotes δ phase precipitation and affects precipitate morphology [30,31]. After 136 rolling, the alloys show a significant loss in ductility due to the formation of numerous δ precipitates along the boundaries (the rolling direction, Figure 7a). Therefore, an annealing process at 1000 °C for 30 min was conducted to promote the complete recrystallization and the partial dissolution of the bulky δ precipitates. It has been proved that the δ phase begins to dissolve at 980 °C and completely dissolves at 1038 °C [32]. Thus, the ductility can be improved while maintaining a relative high strength. The strain-hardening ability is expected to improve greatly by promoting grain refinement and formation of the δ phase (Figure 7b). During the aging process, a large number of fine and coherent particles occur in the alloys, which have been identified as γ′ precipitates (Figure 5 and Figure 7c). Compared to the STed alloy, the ultimate tensile strength, yield strength, and elongation of aged alloys increase by 732 MPa, 518 MPa, and 8%, respectively. This also illustrates that although δ precipitates accumulate at grain boundaries [33,34] and may reduce the ductility, the formation of recrystallized grains and γ′ precipitates can effectively reduce this damage by using optimal heat treatment. 

### 4.2. Strengthening Mechanisms

To investigate the contribution of various strengthening mechanisms to the yield strength of the as-prepared Ni-based superalloys, the strengthening effects of the solid solution, grain boundaries, dislocations, and precipitates are quantified. The yield strength of the Ni-based superalloys is calculated using the following equation [35]:(1)$$\sigma_{y}=\sigma_{0}+\sigma_{ss}+\sigma_{d}+\sigma_{ps}+\sigma_{g}$$
where $\sigma_{y}$ is the yield strength of the material, σ_0_ is a material constant related to the resistance of the dots to dislocation movement, and $\sigma_{ss}$, $\sigma_{d}$, and $\sigma_{ps}$, $\sigma_{g}$ denote the contribution of solid-solution strengthening, dislocation strengthening, precipitation strengthening, and grain refinement strengthening to the yield strength, respectively.

As can be seen from Figure 2, the recrystallization process is basically completed for the three samples, and therefore, the dislocation density is almost negligible. Meanwhile, the solid solution strengthening is also neglected since the elements Al, Ti, and Nb are the main constituents of the precipitated phases, while the atomic radii of the elements Cr, Co, and Mo do not differ much from those of Ni. $\sigma_{g}$ is calculated by using the classical Hall–Petch relationship [35]:(2)$$\sigma_{g}=k_{y}D^{-1/2}$$
where $k_{y}$ is a constant and *D* is the average grain size of the material. Thus, $\sigma_{g}$ is calculated to be 50 MPa, 41 MPa, and 35 MPa for the A10A, A30A, and A60A alloys, respectively.

From the perspective of the interaction between dislocations and precipitates, the precipitation strengthening can be roughly divided into two types, i.e., shear mechanism and bypass mechanism (Orowan-type). In general, when the average diameter of the precipitates is less than 3.5 nm, it usually follows the shear mechanism [36]. Here, according to the statistical results, the average diameter (*d*) of the γ′ precipitates is 7.4 nm in the A30A sample. Thus, the strength increment ($∆\sigma_{Orowan}$) can be estimated by the Ashby–Orowan equation [37]:(3)$$∆\sigma_{Orowan}=(0.528Gbf^{\frac{1}{2}}/d)ln(d/2b)$$
where *f* is the volume fraction of the precipitates with a value of 3.96% for the A30A alloy. Thus, the calculated $∆\sigma_{Orowan}$ = 750 MPa. 

On the other hand, the yield strength, $\sigma_{0}$, arising from grain boundary strengthening and the lattice friction stress of the fcc matrix, can be evaluated by using the following equation [38,39]:(4)$$\sigma_{0}=\sigma_{y}/[\left(1+0.5f\right)\cdot \left(1+{∆\sigma }_{Orowan}/\sigma_{0}\right)]$$

As a result, the calculated $\sigma_{0}$ is 293 MPa. Hence, the contributions of the above strengthening mechanisms to the yield strength of the three samples are summarized in Figure 8a. It is clear that the sum of the calculated intensities (histograms) is slightly higher than the measured values (red dots). The main reason for this discrepancy may be the difficulty in estimating the coupling relationship of the strength contributions. Overall, the multiple strengthening mechanisms of the novel Ni-based superalloy result in an excellent combination of strength and ductility, showing significant advantages over the reported mechanical performances for superalloys with similar chemical compositions [8,25,40,41,42,43,44,45,46,47,48,49,50] (Figure 8b).

### 4.3. Deformation Mechanisms 

To explore the relationship between the ductility and the microstructure of the as-prepared superalloy, the microstructural features of the fracture surface were observed (Figure 9). For the three specimens, no obvious necking can be seen from the macro fracture surface (Figure 9a–c). Close views of the regions in the squares reveal that there are secondary cracks, small dimples, and quasi-cleavage facets in the three alloys, suggesting mixed characteristics of ductile and brittle fracture (Figure 9(a1)–(c1)). Moreover, small and shallow dimples can be distinctly seen at the fracture surface of the A10A alloy (Figure 9(a1)), suggesting the lowest ductility of the alloy (Figure 6a). For the A30A alloy, the dimples become slightly bigger and deeper (Figure 9(b1)), which is consistent with better ductility (Figure 6a). However, the size and number of dimples further increase on the fracture surface of the A60A alloy, accompanied with a few intergranular secondary cracks (Figure 9(c1)), showing a typical ductile fracture characteristic. Furthermore, it can be clearly observed that the length and width of the secondary cracks are large on the fracture surface of the A10A alloy, while they are small in the A60A alloy. This may be due to the stress concentration around the δ precipitates and grain boundaries during the deformation process (Figure 9(a2)). In general, the cracks might arise from the vicinity of the δ precipitates and then extend along the grain boundaries, as the red arrows indicate. Since there are more δ precipitates and the average grain size is small in the A10A alloy (Figure 2), it is more likely to induce large cracks. As shown in Figure 2 and Figure 5, the banded δ precipitates show a denser distribution along the grain boundaries; hence, the voids can easily join into large cracks due to the stress concentration induced by the δ precipitates. That is to say, the δ phase can act as the source of crack initiation, and the banded δ phase is more densely distributed in the A10A alloy (Figure 2 and Figure 5); hence, it is reasonable to see more secondary cracks in the A10A after fracture. Among the three alloys, the content of the δ phase is the highest while the average grain size is smallest for the A10A alloy, which is conducive to the aggregation, growth, and propagation of the cracks. As a result, the ductility of the A10A is low (Figure 6). With increase in the annealing time, the number of δ precipitations decreases and the grain size increases, and the cracks do not easily meet each other during the growth and propagation of the cracks. Thus, smaller secondary cracks are observed in the A30A and A60A alloys (Figure 9(b1),(b2)–(c1),(c2)). Based on the above discussion, it can be concluded that appropriate annealing duration can produce a suitable number of δ precipitates and the optimal grain size, hence promoting a superior combination of strength and ductility in the A30A alloy. 

The microstructures of the deformed A30A sample were characterized by using TEM, high-resolution TEM (HRTEM), EDS mapping, and fast Fourier transformation (FFT), as shown in Figure 10. Dense dislocation tangles (DT) can be observed in the vicinity of the twin boundaries in the deformed A30A sample, which are mainly related to the blockage of twin boundaries to dislocation movements (Figure 10a). The inset shows the misorientation between the twin and the matrix. The stacking fault energy for the γ matrix of Ni-based superalloys are reported to be in the range of 20–30 mJ·m^−2^ [51]. The values are conducive to promoting twinning deformation under the loading, as reported for austenitic stainless steel [11] and TRIP/TWIP steels [12,19,52]. Hence, the twin boundaries, not only those of the annealing twins (Figure 2 and Figure 3) but also those induced by the twining processing, act as an effective blockage to dislocation movements, enhancing dislocation accumulations [18,19]. As a result, numerous dislocation tangles are found in the vicinity of the twins (Figure 10a). In particular, a close view shows that a high density of nanoparticles, dislocations, and SFs coexist (Figure 10b). A large number of dislocations are pinned by the nanoparticles, forming DTs. To determine the type of nanoparticles, STEM-EDS maps providing a closer view of the local region were produced (Figure 10c). Evidently, these nanoparticles are mainly enriched in the elements Ni, Al, Nb, and Ti, which can be regarded as the γ’ phase (Ni_3_(Al, Ti, Nb)). High-resolution TEM shows the features of the deformation microstructure (Figure 10d). 

Several nanoparticles with sizes of ~5 nm can be distinctly seen in the matrix. Figure 10(d1) is a high-resolution TEM (HRTEM) image of the region in the square d1, which exhibits the morphology of spherical particles with a size of approximately 5 nm. In addition, the spacing of (11$\bar{1}$) for the precipitate is 0.205 nm, while that for the matrix is 0.203 nm, i.e., a coherent interface exists between the precipitate and the matrix. The corresponding SAED patterns reveal that the precipitate is γ′ phase, while the γ′ phase and the matrix follow the misorientation relationships of (0$\bar{1}$1)_γ’_//(1$\bar{1}$1)_γ_ and [011]_γ’_//[011]_γ_ (Figure 10e). During deformation, dislocation is more likely to occur in the γ phase and transfers through the γ/γ′ coherent interface [17]. Numerous γ′ precipitates were detected in the as-prepared alloys (Figure 5d), which can effectively block the dislocation movements, hence causing the pile-ups of dislocations. Meanwhile, the premature failure of the alloy may be inhibited by the pass across the dislocations, hence relieving stress on the coherent interfaces [53]. 

Another noticeable feature is that a few Lomer–Cottrell (L-C) locks are observed near the SFs, as the green circles indicate (Figure 10d). The close view of an L-C lock shows that it forms by the intersection of the SFs and a twin at an angle of about 70.5° while the twin has a thickness of 2 nm (Figure 10(d2)). Indeed, L-C locks associated with stair-rod dislocations have been observed near grain and twin boundaries in the rolled nanocrystalline Ni, which were proposed to be effective barriers to mobile dislocations and result in work hardening [54,55]. Furthermore, the presence of nanotwins in the deformed A30A sample can be seen (Figure 10(d3)), which may be attributed to the stacking faults formed near the deformation twins acting as the twin nuclei for the formation of twins. After atomic diffusion and rearrangement, the stacking faults form deformation twins [7,18]. A localized magnification of the region in the yellow rectangle is shown as Figure 10(d3); evidently, they are twins with the thickness of several nanometers, supported by the corresponding SAED patterns (top inset). The numerous SFs and twins in different stacking directions can intersect each other, which is conducive to improving the yield strength and ultimate tensile strength of the Ni-based superalloy [55,56,57]. The formation of large SFs and mechanical twins occurs more easily in the low SFE metals. The progressive formation of SFs and twins on intersecting planes during deformation means that the obstacles to later glides also increase progressively during deformation. This should produce high strain hardening [12,17]. For example, Yamakov et al. [58] suggested that the activation and propagation of deformation twinning can significantly influence the mechanical behavior of metallic materials: (i) when the grain interiors are practically free of dislocations, it can facilitate the deformation by adding additional slip systems and by promoting the transfer between existing slip systems through dislocation-twin reactions; (ii) once twins are formed they can repel certain types of gliding dislocations and give rise to pile-ups, contributing to extra strain hardening for the material. On the other hand, the twin lamellar structure may be regarded as inherently bimodal [18] because the length scale (softer direction) of the “2D grain” parallel to the twin boundaries is significantly larger than the ultrafine scale down to nanometers in the direction perpendicular to the twin boundaries (harder direction). Dislocations can thus accumulate to form dense DTs, thereby subdividing the twin lamellae. In particular, the twin boundaries in large numbers also serve as the locations where dense dislocations can move and accumulate starting from low levels. In situ nanoindentation shows solid evidence for significant work hardening in nanocrystalline Ni based on sequential loading–unloading cycles. The dislocation density along the twin boundaries increases, and the yield strength increases gradually by 40% [58]. Meanwhile, frequent formation of L-C locks is identified in the grain interior and along twin boundaries, which also serve as effective barriers to dislocations and lead to strain hardening. 

On the other hand, the existence of numerous deformation twins, SFs, and L-C locks can significantly contribute to the strain-hardening behavior in Ni-based superalloy based on the microstructural characteristics of the A30A alloy after tensile deformation (Figure 10). The formation of L-C locks is a dynamic process through the interactions among dislocations, SFs, and nanotwins. In addition, L-C locks are sessile structures occurring besides the Frank partial dislocation. The extended dislocations are essential for the formation of an L-C lock. First, two perfect dislocations with Burgers vectors of a/2 [$10\bar{1}$] and a/2[011] are decomposed into partial dislocations on the slip plane of (111) and ($1\bar{1}\bar{1}$) [59], as follows:(5)$$\frac{a}{2}\left[10\bar{1}\right]=\frac{a}{6}\left[2\bar{1}\bar{1}\right]+\frac{a}{6}\left[11\bar{2}\right]$$
(6)$$\frac{a}{2}\left[011\right]=\frac{a}{6}\left[112\right]+\frac{a}{6}\left[\bar{1}21\right]$$

Subsequently, the extended dislocations decomposed from the perfect dislocations interact with each other to generate a new dislocation:(7)$$\frac{a}{6}\left[\bar{1}21\right]+\frac{a}{6}\left[2\bar{1}\bar{1}\right]\to \frac{a}{6}\left[110\right]$$

Here, the newly formed a/6[110] dislocation is generally sessile on the slip plane in fcc, known as the stair-rod dislocation or L-C lock. The L-C locks can stabilize the SFs through the pinning effect when two extended dislocations meet, thus blocking the movement of other slipped dislocations simultaneously [54,60]. Hence, it can be concluded that the numerous SFs and L-C locks observed in the deformed A30A alloy play a key role for enhancing the strain-hardening capability of the sample during tensile testing. Indeed, this has been proved in a Ni-based GH3536 (Hastelloy X) superalloy deformed at cryogenic temperature [61]. The related deformation process is illustrated in the schematic diagrams shown in Figure 11. Before deformation, dislocation can be scarcely seen inside the grain and at the grain boundaries because of the complete recrystallization and aging treatment, and the microstructure mainly consists of δ precipitates along the grain boundaries and γ’ precipitates in the interior of the grains (Figure 11a). After tensile deformation, the main strengthening and deformation mechanisms are shown in Figure 11b. Based on the TEM characterization to the deformed A30A sample, the deformed microstructure consists of γ’ precipitates, δ precipitates, SFs, nanotwins, and L-C locks (Figure 10). During the tensile loading, the grain boundaries and δ precipitates can effectively obstruct the dislocation movements. A large number of stacking faults and twins in different stacking directions can interact with each other to form the L-C locks, thus leading to distinct strain hardening. It should be pointed out that the contribution of the γ′ precipitates cannot be ignored; their strengthening contribution during deformation is shown in Figure 8a. The results demonstrate that precipitation strengthening is the main mode of strengthening in the Ni-26.6Co-18.4Cr-4.1Mo-2.3Al-0.3Ti-5.4Nb alloy. By introducing multiple strengthening/deformation mechanisms, the alloy exhibits a superior combination of strength and ductility (Figure 6).

## 5. Conclusions

In this study, a Ni-based superalloy with excellent mechanical properties was prepared via rolling, annealing, and aging processes. Subsequently, the corresponding microstructural evolution, mechanical properties, and deformation mechanisms were explored in detail. The main conclusions that can be drawn are as follows:
(1)At room temperature, the as-prepared A30A alloy shows an ultrahigh yield strength of 1093 MPa, ultimate tensile strength of 1561 MPa, and good ductility of 29%. Compared with the STed alloy, the yield strength and ultimate tensile strength of the A30A alloy increase by approximately 82% and 85%, respectively. In particular, the ductility significantly increases from 23% to 29%.(2)At 700 °C, the strengths of the specimens significantly decrease by about 200 MPa in comparison with the values tested at room temperature. The A30A alloy exhibits the highest yield and ultimate tensile strengths of 833 MPa and 1024 MPa, while retaining a good ductility of 6.3%. The main reason for these results can be related to the fact that the local deformation is enhanced, and the homogenous deformation is retarded accordingly at elevated temperature. (3)The main reinforcing phase of the as-prepared Ni-based superalloy is the γ’ precipitates, which have coherent interfaces with the fcc matrix. The enhanced precipitation of the nanosized γ′ particles by heat treatment can reduce the damage to ductility caused by the bulky δ precipitates. (4)During deformation, the progressive formation of SFs and twins on intersecting planes means that the obstacles to later glides also increase progressively, thus producing high strain-hardening. Consequently, the A30A alloy shows simultaneous increments in strength and ductility compared with the A10A and A60A alloys. The nanoscale twins, dense nanosized γ′ particles, SFs, dislocation networks, and L-C locks are responsible for the ultrahigh strength of the A30A alloy.


## Figures and Tables

**Figure 1 nanomaterials-14-01296-f001:**
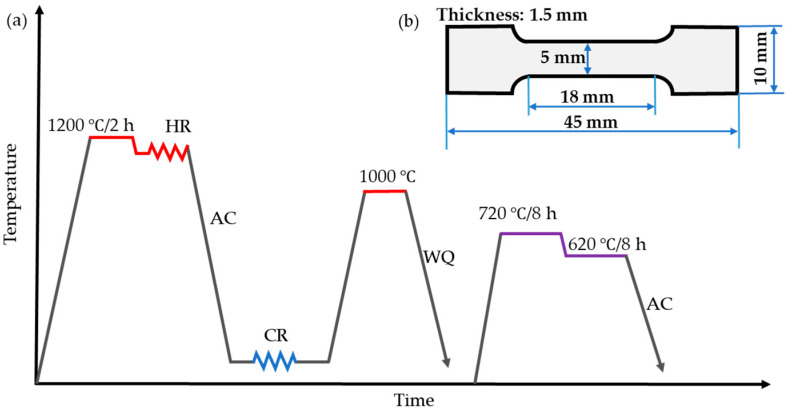
(**a**) Diagram illustrating the Ni-based alloy treatment process, and (**b**) the dimensions of dog-bone shaped specimens for the uniaxial tensile testing.

**Figure 2 nanomaterials-14-01296-f002:**
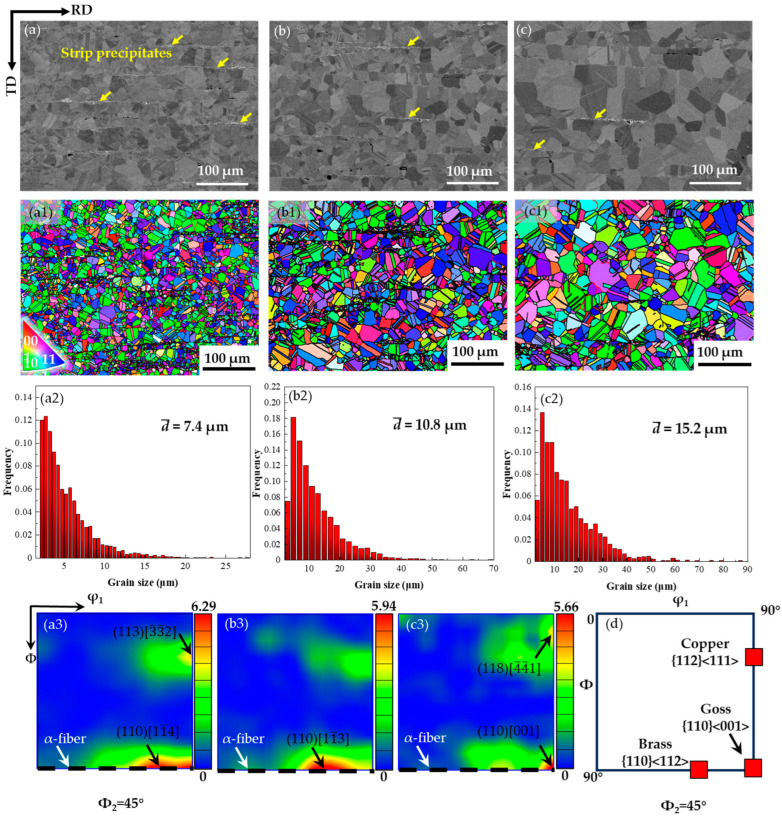
(**a**–**c**) SEM images show the precipitates along the bands, (**a1**–**c1**) EBSD IPF maps revealing the morphology of grains, (**a2**–**c2**) grain size distribution histograms, and (**a3**–**c3**) the related orientation distribution function (ODF) sections of the samples A10A, A30A, and A60A, respectively. (**d**) φ_2_ = 45° ODF sections showing the idea orientations. Arrows in yellow, white and black represent strip precipitates, α-fiber and strongest texture, respectively.

**Figure 3 nanomaterials-14-01296-f003:**
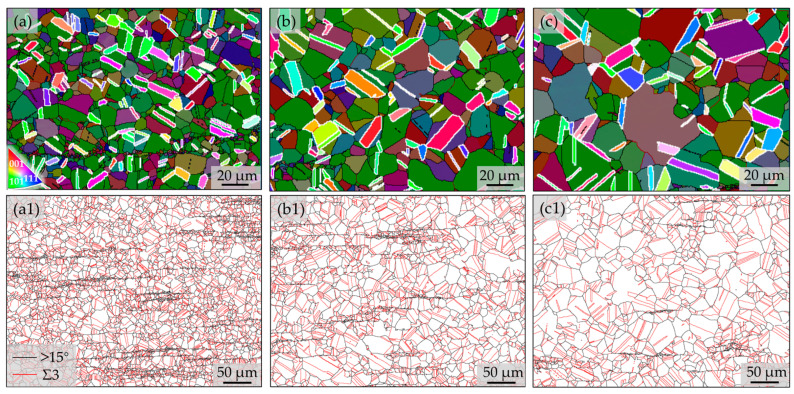
EBSD IPF maps highlighting the distributions of twins and grains (**a**–**c**), and the corresponding images, with black and red lines representing high-angle boundaries with misorientation over 15° and Σ3 twin boundaries, respectively. (**a**,**a1**) A10A, (**b**,**b1**) A30A, (**c**,**c1**) A60A.

**Figure 4 nanomaterials-14-01296-f004:**
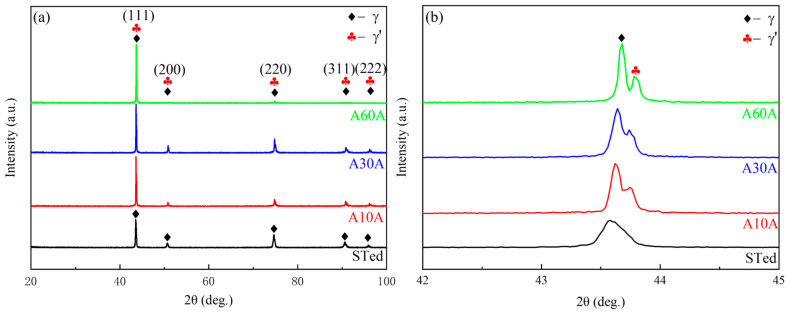
XRD patterns of the STed and A10A, A30A, A60A samples (**a**), together with the magnified views of the (111) peaks (**b**).

**Figure 5 nanomaterials-14-01296-f005:**
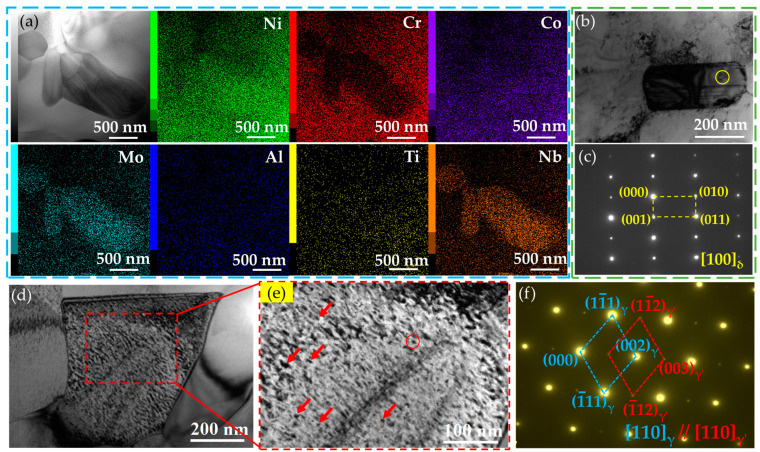
TEM image and the corresponding EDS maps showing the morphologies and element distributions of the precipitates in the A30A alloy (**a**). (**b**) The morphology of a rectangle precipitate at the grain boundary, and (**c**) the SAED patterns of the yellow circle in (**b**). Dense nanosized particles in a grain (**d**,**e**) and the related SAED patterns of the red circle in (**f**). The red arrows point out the nanoprecipitates.

**Figure 6 nanomaterials-14-01296-f006:**
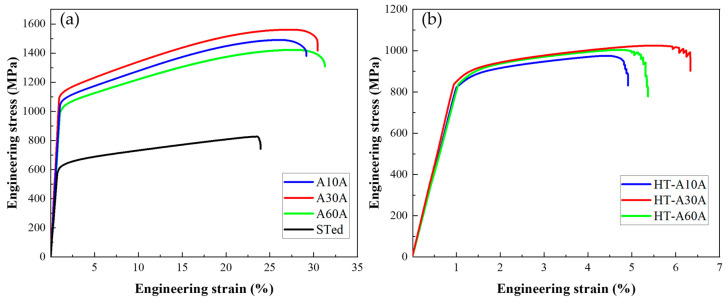
Engineering stress–strain curves of the as-prepared three alloys tested at (**a**) 25 °C and (**b**) 700 °C.

**Figure 7 nanomaterials-14-01296-f007:**
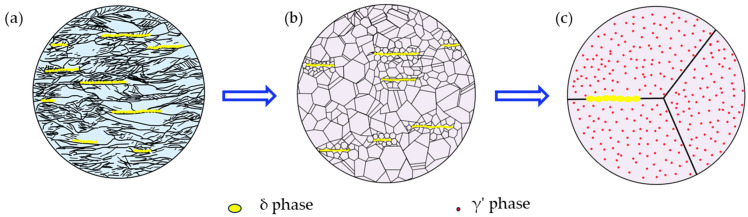
Microstructural evolution during the processing: (**a**) cold-rolling, (**b**) annealing, (**c**) aging.

**Figure 8 nanomaterials-14-01296-f008:**
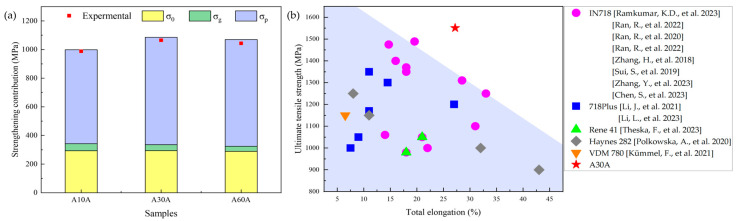
(**a**) The contributions of different strengthening mechanisms for different samples of Ni-based alloy; (**b**) the mechanical properties of Ni-based alloy compared with others at RT [8,25,40,41,42,43,44,45,46,47,48,49,50].

**Figure 9 nanomaterials-14-01296-f009:**
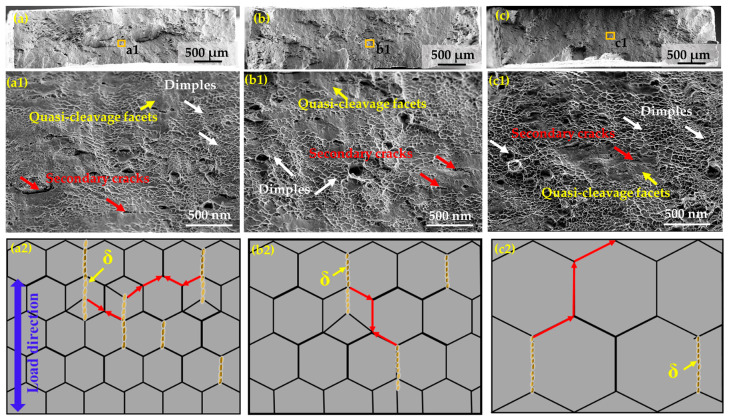
SEM images showing the fracture morphologies and the corresponding schematic diagrams exhibit potential propagation routes of cracks in the three alloys. (**a**–**a2**) A10A, (**b**–**b2**) A30A, and (**c**–**c2**) A60A. Arrows in white, red and yellow represent dimples, secondary cracks and quasi-cleavage facets, respectively.

**Figure 10 nanomaterials-14-01296-f010:**
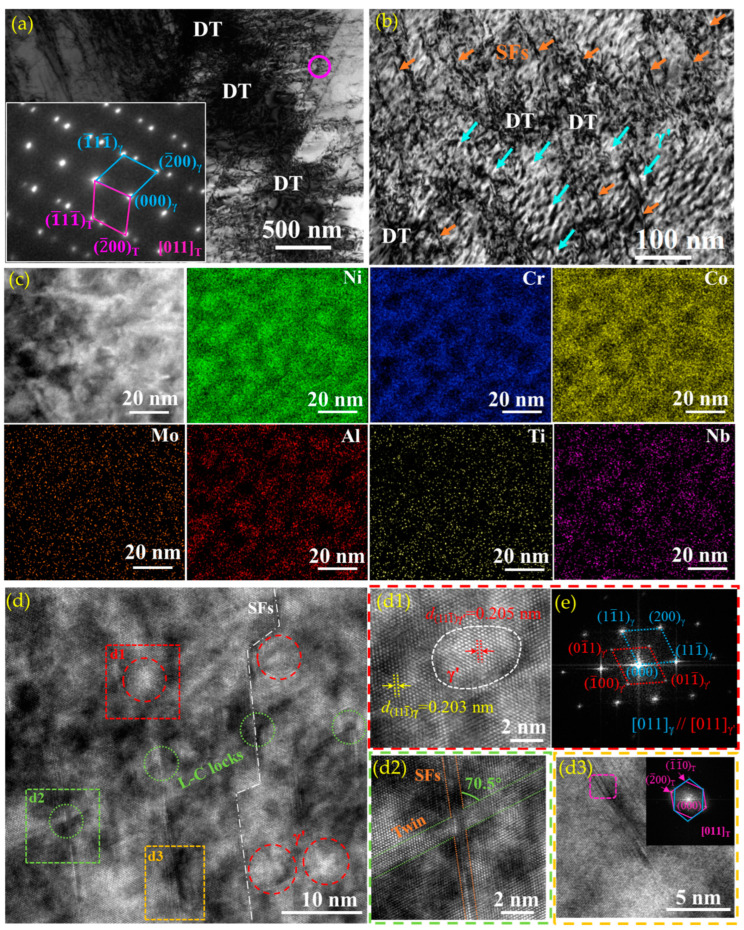
(**a**) TEM images of A30A sample true-strained to 20% and corresponding SAED pattern of the pink circle, TEM images of the tensile fractured A30A sample: (**b**) TEM image, (**c**) STEM image and the corresponding EDS mapping showing the distributions of Ni, Cr, Co, Mo, Al, Ti, and Nb, respectively, (**d**) HRTEM image, local enlargements of (**d**) showing γ′ precipitate (**d1**), L-C lock (**d2**), HRTEM and inset FFT image showing the fine structure and orientation of deformation twins (**d3**), FFT image of (**d1**) showing the orientation of γ′ precipitates (**e**). Arrows in orange and blue represent SFs and γ’, respectively.

**Figure 11 nanomaterials-14-01296-f011:**
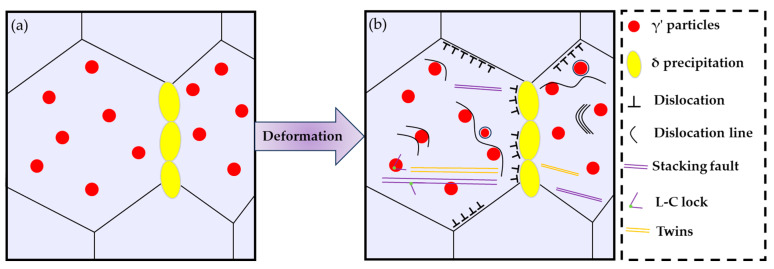
Schematic illustrations showing the deformation mechanisms of the as-prepared Ni-based superalloy under tensile testing at room temperature. (**a**) Before deformation, (**b**) after deformation.

## Data Availability

All data used in this study are available upon request from the corresponding author.

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
