# Peer review of "Enhancing Strength and Ductility of a Ni-26.6Co-18.4Cr-4.1Mo-2.3Al-0.3Ti-5.4Nb Alloy via Nanosized Precipitations, Stacking Faults, and Nanotwins"

_nanomaterials, 2024, doi:10.3390/nano14151296_

Round 1

Reviewer 1 Report

Comments and Suggestions for Authors

The paper entitled " Enhancing strength and ductility of a Ni-26.6Co-18.4Cr-4.1Mo-2 2.3Al-0.3Ti-5.4Nb alloy via nanosized precipitations, stacking 3 faults and nanotwins” by Zhang et al. is an original paper concerning superalloy showing realy interesting properties, ultrahigh yield strength, ultimate tensile strength and ductility. And this is a strong point of this manuscript – the topic is important. The manuscript presents many results obtained by different techniques. The paper is based on the newest references and the discussion is interesting. However, there are some problems indicated in details below. Therefore, I would recommend major revision.

Problems:

Sometimes you use et al instead et al. Please verify.

Why did you selected SiC sandpaper of different grades for sample preparation. Did you test importance of this grade?

Line 134: “boundaries significantly increase from 46% to 53%” Why this increase by 7% is significant and an increase by 5% (from 14 to 19% given one line above) is not significant?

Line 136: there a b c probably coming from a figure given on the next page.

PXRD section I agree that those signals perfectly fits nickel phase. However, can you comment a decrease in intensity of many reflections after annealing. It indicate that number of grains oriented in this way decreases significantly and I would expect some ordering of the structure and texture effect. You wrote that it was not observed (line 116).

How important is a shift of (111) observed on Fig. 4b. It reaches ca. 0.25 2theta.

Line 155: “…have conducted…” and it should be “,,,have been conducted…”

Line 168: (a) probably coming from any figure. It should be removed

Line 170-171: 0.213 and 0.21. Is it important this difference? What is the error of such parameter determination? You indicate 0.21, so it means that it corresponds to 0.210 or the error allows for giving only two numbers and hence this number is limited to 0.21 nm.

Fig. 8. Capture is incomplete. There are no information about (f) and also green rectangle around (d2) and (f) should be limited only to (d2). Also (e) is not HRTEM but SAED image (it is confirmed by text in 256-257 lines).

Line 319: “…phase is the most while…” this sentence is unfinished (the most WHAT)

Line 327: (a) which is not related to any figure

Line 335: “The formation of L-C locks of is…” Why “of”?

Line 337: “The extended dislocations is” should be “The extended dislocations are”

Line 348: “Hence, it is can be concluded” should be “Hence, it can be concluded”

Comments on the Quality of English Language

some changes are needed

Author Response

Dear Editor,

Thank you very much for your correspondence on our manuscript Nanomaterials-3104658 “Enhancing strength and ductility of a Ni-26.6Co-18.4Cr-4.1Mo-2.3Al-0.3Ti-5.4Nb alloy via nanosized precipitations, stacking faults and nanotwins”.

We revised the manuscript according to the reviewers’ comments and highlighted the changes in red font. We think the revision has significantly enhanced the quality of the manuscript. Thank you!

Reviewer #1: The paper entitled "Enhancing strength and ductility of a Ni-26.6Co-18.4Cr-4.1Mo-2.3Al-0.3Ti-5.4Nb alloy via nanosized precipitations, stacking faults and nanotwins” by Zhang et al. is an original paper concerning superalloy showing really interesting properties, ultrahigh yield strength, ultimate tensile strength and ductility. And this is a strong point of this manuscript – the topic is important. The manuscript presents many results obtained by different techniques. The paper is based on the newest references and the discussion is interesting. However, there are some problems indicated in details below. Therefore, I would recommend major revision.

We appreciate the reviewer’s comments which surely help with the improvement of the manuscript. The detailed responses are described as follows:

Comments 1: Sometimes you use et al instead et al. Please verify.

Response 1: Thanks! The corrections have been made in lines 37 and 356.

Comments 2: Why did you selected SiC sandpaper of different grades for sample preparation. Did you test importance of this grade?

Response 2: Thanks for the reminding! The samples used for XRD testing were finely ground. Indeed, it is unnecessary to list so many grades. We have corrected this and highlighted the relevant revisions in red in lines 92-93.

Comments 3: Line 134: “boundaries significantly increase from 46% to 53%” Why this increase by 7% is significant and an increase by 5% (from 14 to 19% given one line above) is not significant?

Response 3: Thanks! The change in twins with increasing annealing time is illustrated here. Two metrics are used here, the increase of both the percentage of twin boundaries to total boundaries and the area fraction of twins are intended to quantitative statistics. The increasement of the two indicators is similar, so the manuscript has been modified in line 135.

Comments 4: Line 136: there a b c probably coming from a figure given on the next page.

Response 4: Sorry! We will move them to Fig. 3 of page 6.

Comments 5: PXRD section I agree that those signals perfectly fits nickel phase. However, can you comment a decrease in intensity of many reflections after annealing. It indicate that number of grains oriented in this way decreases significantly and I would expect some ordering of the structure and texture effect. You wrote that it was not observed (line 116).

Response 5: Special thanks for the constructive suggestion! In this version, we have added orientation distribution function sections to Fig. 2 at page 5. The supplemented contents are highlighted in red in lines 119-129 and lines 146-148.

Comments 6: How important is a shift of (111) observed on Fig. 4b. It reaches ca. 0.25 2theta.

Response 6: Thanks for the valuable suggestion! In this version, we add the related descriptions in the lines 150-152 in the revised manuscript. The specific supplements are as follow:

Compared with Sted alloy, a shift of (111) diffraction peak of aging alloys (A10A, A30A, A60A) is observed in Fig. 4b, which indicates that the lattice constants of the alloys become smaller after aging treatment. The phenomenon is caused by the precipitation of the γ’ phase on the matrix.

Comments 7: Line 155: “…have conducted…” and it should be “,,,have been conducted…”

Response 7: Sorry for the mistake! The correction has been made in line 158.

Comments 8: Line 168: (a) probably coming from any figure. It should be removed

Response 8: Sorry! The correction has been made on page 7.

Comments 9: Line 170-171: 0.213 and 0.21. Is it important this difference? What is the error of such parameter determination? You indicate 0.21, so it means that it corresponds to 0.210 or the error allows for giving only two numbers and hence this number is limited to 0.21 nm.

Response 9: Thanks! It is indeed difficult to make an accurate distinction for the spacing of the matrix and precipitates from SAED. Therefore, we removed the relevant description.

Comments 10: Fig. 8. Capture is incomplete. There are no information about (f) and also green rectangle around (d2) and (f) should be limited only to (d2). Also (e) is not HRTEM but SAED image (it is confirmed by text in 256-257 lines).

Response 10: Thanks for the important suggestion! We have modified Fig. 10 (original Fig. 8) and the corresponding caption on page 16.

Comments 11: Line 319: “…phase is the most while…” this sentence is unfinished (the most WHAT)

Response 11: We are so sorry for the mistake! It should be " the content of δ phase is the highest ", and corresponding correction has been made in line 301 of page 12.

Comments 12: Line 327: (a) which is not related to any figure

Response 12: Sorry! We will delete it in this version.

Comments 13: Line 335: “The formation of L-C locks of is…” Why “of”?

Response 13: Sorry for the mistake. The correction has been made in line 373.

Comments 14: Line 337: “The extended dislocations is” should be “The extended dislocations are”

Response 14: We are so sorry for the mistakes! We have modified it in line 375.

Comments 15: Line 348: “Hence, it is can be concluded” should be “Hence, it can be concluded”

Response 15: Thanks! We will modify it in the revised manuscript in line 386.

Thank you very much for kindly considering this revised version!

Yours sincerely,

Professor: Yongfeng Shen, on behalf of all co-authors.

Reviewer 2 Report

Comments and Suggestions for Authors

This study developed Ni-based superalloys with a high Co content to decrease SF energy. Dense dislocation tangles, nano twins, and SF and LC locks are the reasons for improving strength. The experiments are very consistent and the results are very clear. This paper is acceptable after minor revision as follows.

1.     Line 41: single-phase Ni-based alloys. The authors indicate gamma’ precipitates. Then, “single phase” may be “single-crystal”. Check it and revise to appropriate word.

2.     Line 113: “Strip-like precipitates after heat treatment.” When do these precipitates form? During heat treatment? Then, why do the precipitates decrease with the increase of annealing time? The precipitates form before heat treatment; I understand the author's claim.

3.     Line 142: no explanation of “STed”. Explain this.

4.     Line 164-165: In which stage does the delta phase form? During HR? please clarify it.

5.     In line 168, the letter (a) appears. Delete it.

6.     Line 205: Authors indicate the yield strength of 690 MPa at 649C for Ni-based superalloy. Please show which Ni-base super alloys you compared. 

7.      Discussion, 4.1 Microstructural evolution. 4.2 and 4.3 indicate different heat treatment conditions. Then, 4.2 and 4.3 should be in 4.1; that is, 4.1.2 As-prepared samples, 4.1.3 After tensile test

8.     Line 416: delete 1093 MPa and 1561 MPa

Author Response

Dear Editor,

Thank you very much for your correspondence on our manuscript Nanomaterials-3104658 “Enhancing strength and ductility of a Ni-26.6Co-18.4Cr-4.1Mo-2.3Al-0.3Ti-5.4Nb alloy via nanosized precipitations, stacking faults and nanotwins”.

We revised the manuscript according to the reviewers’ comments and highlighted the changes in red font. We think the revision has significantly enhanced the quality of the manuscript. Thank you!

Reviewer #2: This study developed Ni-based superalloys with a high Co content to decrease SF energy. Dense dislocation tangles, nano twins, and SF and LC locks are the reasons for improving strength. The experiments are very consistent and the results are very clear. This paper is acceptable after minor revision as follows.

We appreciate the reviewer for the comments and suggestions to improve the quality of our manuscript. The point-to-point responses are detailed below.

Comments 1: Line 41: single-phase Ni-based alloys. The authors indicate gamma’ precipitates. Then, “single phase” may be “single-crystal”. Check it and revise to appropriate word.

Response 1: Thanks! We have modified the related descriptions in line 39.

Comments 2: Line 113: “Strip-like precipitates after heat treatment.” When do these precipitates form? During heat treatment? Then, why do the precipitates decrease with the increase of annealing time? The precipitates form before heat treatment; I understand the author's claim.

Response 2: Thanks to the reviewer for pointing this important question. The strip-like precipitates form before heat treatment. “Strip-like precipitates after heat treatment” is our misstatement to cause a misunderstanding. We have modified it on page 4 in the manuscript.

Comments 3: Line 142: no explanation of “STed”. Explain this.

Response 3: Thanks! “STed” means that the cast state alloy has been subjected to the solution-treated. We added this explanation in lines 71-72 of page 3.

Comments 4: Line 164-165: In which stage does the delta phase form? During HR? please clarify it.

Response 4: Thank you very much for your suggestion! Previous studies have shown that cold rolling promotes the precipitation of δ phase [30-31], and a similar phenomenon is found in this study. Therefore, δ phase is produced after cold rolling. We added the relevant explanation in lines 223-224.

Comments 5: In line 168, the letter (a) appears. Delete it.

Response 5: Thanks! The correction has been made on page 7.

Comments 6: Line 205: Authors indicate the yield strength of 690 MPa at 649C for Ni-based superalloy. Please show which Ni-base super alloys you compared.

Response 6: Thank you very much for your useful suggestion! we will modify it in line 204 of page 9.

Comments 7: Discussion, 4.1 Microstructural evolution. 4.2 and 4.3 indicate different heat treatment conditions. Then, 4.2 and 4.3 should be in 4.1; that is, 4.1.2 As-prepared samples, 4.1.3 After tensile test

Response 7: Thanks for your valuable suggestion! We have restructured the discussion section following your suggestion. The microstructural evolution after rolling and aging treatment is given as section 4.1, and the strengthening mechanisms of samples after aging treatment are given as section 4.2. Besides, the morphology of the alloy after tensile fracture is integrated into the original 4.2 (Deformation Mechanisms) and given as section 4.3.

Comments 8: Line 416: delete 1093 MPa and 1561 MPa

Response 8: Sorry for the errors! We have removed from the main text in line 419.

Thank you very much for kindly considering this revised version!

Yours sincerely,

Professor: Yongfeng Shen, on behalf of all co-authors.

Round 2

Reviewer 1 Report

Comments and Suggestions for Authors

After the changes introduced by the Authors I am pleased to recommend this paper for publication in Nanomaterials.